# PASS: Predictive Auto-Scaling System for Large-scale Enterprise Web Applications

## ABSTRACT

We confront two challenges in the management of a vast and diverse array of online web applications deployed on enterprise-grade auto-scaling infrastructure, primarily focused on ensuring Quality of Service (QoS) for large-scale applications and optimizing resource costs. Firstly, reacting to increased load with a response-based approach can temporarily degrade QoS because many web applications need a few minutes to warm up. Therefore, precise workload prediction is critical for predictive scaling. However, our analysis of real-world applications underscores the substantial challenges arising from the limited precision and robustness of existing single prediction algorithms in the context of predictive auto-scaling. Secondly, guaranteeing the QoS of online applications within a cost-effective structure is crucial, as it is inherently linked to corporate profitability. Nevertheless, our study shows that mainstream auto-scaling methods exhibit various limitations, either being unsuitable for online environments or inadequately ensuring QoS.

To address these issues, we introduce **PASS**, a **P**redictive **A**uto-**S**caling **S**ystem tailored for large-scale online web applications in enterprise settings. Our highly robust and accurate prediction framework dynamically integrates and calibrates appropriate prediction algorithms based on the unique characteristics of each application to effectively manage workload diversity. We further establish a performance model derived from online historical logs, enhancing auto-scaling to ensure diverse QoS without adverse impacts on online applications. Additionally, we implement a reactive strategy grounded in queuing theory to promptly address QoS violations resulting from inaccurate predictions or unexpected events. Across a wide spectrum of applications and real-world workloads, PASS outperforms state-of-the-art methods, achieving higher workload prediction accuracy and a superior QoS guarantee rate with less resource cost.

## KEYWORDS

auto-scaling, workload prediction, quality of service, performance model, cloud computing

## 1 INTRODUCTION

In recent years, the rapid growth of cloud computing has prompted an increasing number of enterprises to host their web applications on public clouds, such as Amazon EC2 [1] and Windows Azure [2], or on private clouds managed by frameworks like VMware Cloud [4], Mesos [19], and Kubernetes [8]. Auto-scaling has become a key facilitator for efficiently allocating and releasing resources in response to workload fluctuations[7], leading to substantial reductions in operational and management costs [37, 38].

We conducted a comprehensive assessment and analysis of auto-scaling within our large-scale internet enterprise's private cloud platform. Our study encompassed over 200 services from eight business departments within the enterprise. The findings demonstrated that the methods employed within the enterprise and some other state-of-the-art (SOTA) techniques either inadequately ensured

Quality of Service (QoS) or led to excessive resource redundancy. This stems from the following two challenges:

**The first challenge revolves around the necessity for precise workload forecasting technology.** As enterprises cut costs and boost efficiency, many departments refrain from keeping excess resources to handle peak workloads. However, this reliance on dynamic resource scaling presents challenges. Reactively scaling to ensure QoS during workload surges is complicated by the warm-up time required for starting program instances. This delay can temporarily reduce QoS, highlighting the importance of accurate load prediction technology for maintaining high-quality service. Furthermore, we randomly collected Query Per Second (QPS) time series data from 200+ enterprise web services and found that existing SOTA prediction algorithms either lack precision or robustness, making them incapable of capturing critical traffic mutation features[1] in a timely manner (see Section 2.1). This results in inadequate predictive performance, rendering them unable to serve downstream auto-scaling methods to guarantee QoS.

**The second challenge is lack of an accurate and cost-effective performance model to guarantee QoS for online applications.** Performance models are used to estimate required resources based on levels of workload. Ideally, an accurate model can be constructed by sampling in an offline environment. However, the large scale and extensive range of enterprise applications make it expensive and challenging to replicate an offline environment identical to the online setup. Furthermore, the direct link between the service quality of online applications and corporate profits makes it a priority for application owners. As a consequence, application profiling [14, 36, 43] and AI-driven auto-scaling methods like reinforcement learning [12, 39] are not universally applicable due to their potential adverse impact on the online applications. Moreover, the auto-scaling methods commonly employed by application owners, such as threshold-based rules [28], target tracking [3], and queuing theory [24], often struggle to effectively guarantee the QoS of a wide range of web applications due to their simple or inaccurate performance model (see Section 2.2).

In this paper, we propose **PASS**, a **P**redictive **A**uto-**S**caling **S**ystem designed for large-scale online web applications in enterprise environments. **To address the first challenge,** we have developed a predictive algorithm specialized in forecasting QPS time series for enterprise-level applications. This algorithm integrates various sub-algorithms customized to the specific features of an application's QPS series, thereby enhancing prediction accuracy. **To tackle the second challenge,** we have constructed a performance model based on historical online logs to guarantee diverse QoS for applications without compromising online performance. PASS utilizes the predicted QPS to access the performance model and determine the number of instances, ensuring that scaling out occurs prior to the warm-up time. Furthermore, the system continuously monitors

---

[1]the mutation features are especially prevalent and critical in enterprise-level applications. This is due to a substantial influx of QPS during morning, noon, and evening peak hours, leading to a considerable and undeniable impact.

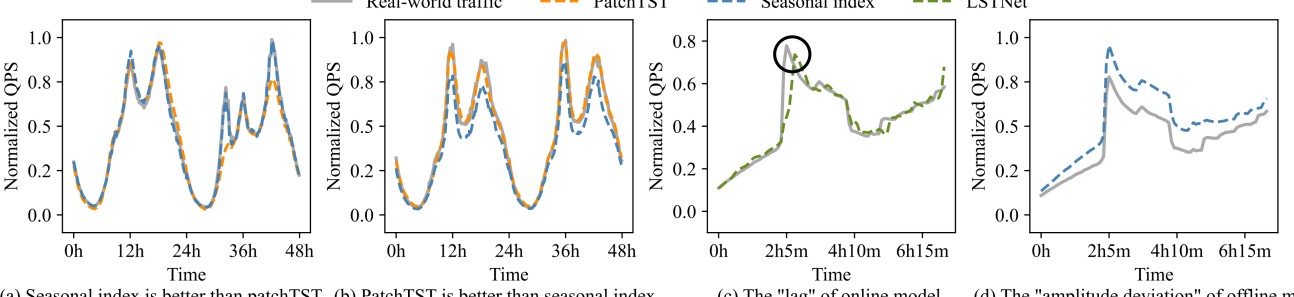

(a) Seasonal index is better than patchTST.  (b) PatchTST is better than seasonal index.  (c) The "lag" of online model.  (d) The "amplitude deviation" of offline model.

**Figure 1: Comparing the accuracy and robustness of various prediction algorithms.**

the QoS metrics of the application in real time. Upon detecting a violation, PASS promptly revises the prediction QPS and scales out to the appropriate number of instances, with the aim of minimizing the duration of the QoS violation. Employing ELPA, PASS effectively lowers average resource consumption by 8.91% compared to the SOTA method while achieving a notable improvement in QoS assurance, with a rate increase of up to 22.64%.

We make four primary contributions in this paper. Firstly, we conducted real-system testing of mainstream prediction algorithms[2] and auto-scaling methods for enterprise-level services. Furthermore, we conducted a thorough evaluation of existing methods, highlighting the limitations that were previously overlooked in the context of large-scale web applications auto-scaling (see Section 2). Secondly, we've developed an **E**nsemble **L**earning-based **P**rediction **A**lgorithm named **ELPA**. This algorithm leverages a specialized combination of sub-algorithms tailored to the time series QPS characteristics of various enterprise-level applications, resulting in improved prediction accuracy (see Section 3.1). Thirdly, we introduced PASS, a predictive auto-scaling system for large-scale web applications. PASS prioritizes QoS while remaining cost-effective, and doesn't need profiling or negatively impact the applications (see Section 3.2 and 3.3). Finally, we evaluated the performance of ELPA and PASS using a diverse range of enterprise web applications, demonstrating that our methods outperform SOTA approaches in the majority of cases (see Section 4).

The rest of the paper is organized as follows. Section 2 evaluates mainstream prediction algorithms and elastic scaling methods in our enterprise system scenarios and provides an overview of the challenges associated with auto-scaling web applications in enterprise environments. Section 3 outlines our auto-scaling framework's process, as well as the design of the prediction and performance models. Section 4 presents the experimental results obtained from our enterprise systems. Section 5 discusses related work, and Section 6 serves as the conclusion of the paper.

## 2 REAL-SYSTEM INVESTIGATION

We first evaluate and analyze the mainstream prediction algorithm (Section 2.1) and auto-scaling method (Section 2.2) on real systems and applications. And then in Section 2.3, we summarize the challenges of large-scale web application auto-scaling in our enterprise.

### 2.1 Prediction Algorithm Analysis

we employed a stratified sampling method across 10 thousand services operating in real business scenarios across various business

groups within our company. This allowed us to apply the Pearson approach [17] (see Appendix A.1 for details) to detect periodicity within the data. Our findings indicate that 92.80% of the applications exhibit strong periodic patterns, 4.55% display weaker periodicity, and only 2.65% show no discernible periodicity.

**Observation 1: For the majority of applications, the real-world load can be reliably forecasted using models. This is primarily because the time-series load data for most businesses remains consistent and predictable.**

By conducting experimental comparisons, we rigorously evaluated the performance of a range of prediction algorithms commonly employed in both industry and academia, spanning various application types. We subjected 225 real-world applications to testing using their respective time-series data[3]. Our observations reveal that the most effective prediction algorithm depends on the specific characteristics of business traffic, emphasizing the absence of a universal solution. As depicted in Figure 1(a) and (b), we illustrate the predictive performance of seasonal index and patchTST using the traffic time-series data from three representative applications. In Figure 1(a), patchTST significantly outperforms the seasonal index for application 1, while in Figure 1(b), the seasonal index proves more effective than patchTST for application 2. These results underscore the variability in algorithm performance, highlighting the need to select the most suitable algorithm for specific time-series traffic characteristics.

**Observation 2: Owing to the inherent constraints of prediction algorithms, there is no single algorithm capable of consistently providing optimal predictive performance for all categories of time-series data characteristics. Diverse prediction algorithms prove more effective for specific types of business traffic forecasts.**

Prediction algorithms are commonly categorized into two technical approaches: online prediction and offline prediction. Online prediction models utilize real-time traffic (time series) data to forecast values in the near future. This continuous intake of the latest time-series data typically leads to superior predictive performance[4]. Conversely, offline prediction models can directly predict future traffic values without real-time data input, often relying on historical data. However, offline models, which lack the latest data, may not achieve the same level of predictive accuracy.

Offline prediction has its merits. In scenarios where rapid response is critical, timely and precise (robustness) predictions are

---

[2]We analyzed the outcomes of the prediction algorithms across 200+ services spanning 8 different business departments within the enterprise.

[3]Specific details regarding the evaluated algorithms can be found in Section 4.1.
[4]This process is akin to a sliding window, with the latest real-time traffic data continuously entering the model, producing sequential predictions for future traffic values.

|  | QoS Guarantee Rate | | Resource Cost |
|---|---|---|---|
|  | avg RT | TP999 RT | |
| Threshold based rules | 62.16% | 54.05% | 85 |
| Target tracking | 74.29% | 68.57% | 112 |
| Queuing theory | 88.57% | 80% | 92 |

**Table 1: QoS guarantee rate and resource cost of three common auto-scaling methods.**

more critical than accuracy. For example, in cases of periodic "mutation features," online models may exhibit prediction lags (highlighted in a black circle in Figure 1(c)). This lag effect will persist for a while, which we refer to as the "dirty interval". On the contrary, offline models effectively capture this information, facilitating timely auto-scaling actions (see Figure 1(d)). However, offline models can encounter issues like "amplitude deviation," where the shape of the feature is accurately represented but there's a discrepancy in its absolute value (see Figure 1(d)). Note that the "mutation features" are especially prevalent in enterprise-level applications. This is due to a substantial influx of traffic during morning, noon, and evening peak hours, leading to a considerable and undeniable impact.

**Observation 3: Online methods offer superior average accuracy and stable predictive performance but struggle with "mutation features." In contrast, offline approaches can capture "mutation features" but suffer from significant amplitude deviations.**

## 2.2 Auto-scaling Methods Analysis

The auto-scaling methods of web applications can be briefly divided into the following categories [37, 38]:

**1) Threshold based rules** [1, 18, 21]. Auto-scaling is performed based on a series of rules containing thresholds. Taking CPU resource utilization as an example, when the resource utilization exceeds the upper threshold, resources are increased, and conversely, resources are decreased when the utilization falls below the lower threshold. Threshold parameters are generally set empirically by the application owner.

**2) Target tracking** [3]. A control theory [33] method that maintains a certain metric (such as CPU resource utilization) within a specific range. When the actual resource utilization is not within the set range, the number of instances that need to be scaled is automatically calculated based on the current status. For example, if the current average CPU resource utilization of 10 instances is 80% and the target value is 50%, then the instances need to be scaled to $80\% * 10/50\% = 16$ instances.

**3) Queuing theory** [5, 20]. It estimates performance metrics and the waiting time of requests based on queuing theory models. The common $M/M/s$ model indicates that the time for request arrival and processing is exponentially distributed, and a total of $s$ servers process it in parallel. AHPA [49] also uses Queuing theory as a performance model.

**4) Reinforcement learning** [13, 42]. RL is a widely used method, where the auto-scaler acts as an agent and interacts with the environment, receiving reward feedback after each action. It establishes a mapping model between states and actions through trial and error [12, 39].

The methods used by most application owners are simple or direct, including: threshold based rules, target tracking, and queuing theory. RL and other AI related methods are not in the set because they may affect the performance of online applications. In order to

verify the QoS guarantee degree of the commonly used methods, we selected a representative back-end service that provides data support and tested the QoS guarantee rate under two kinds of QoS. Since our main purpose is to verify the degree of QoS guarantee, we used a ladder-increasing workload. Table 1 shows the results of QoS guarantee rate and resource cost. The QoS guarantee rate is calculated by dividing the duration of QoS guarantee by the total time length. The resource cost is determined by the integral of the number of instances over time(minute).

**Observation 4: The auto-scaling methods widely used by web application owners do not effectively guarantee QoS, especially when the QoS requirement includes tail latency.** Compared with QoS of average Response Time (RT), the QoS guarantee rates of the three methods are significantly reduced when the QoS requirement is TP999 tail latency. However, most web applications require tail latency QoS, which makes widely used methods even less effective, either violating QoS often or wasting resource.

**Observation 5: The performance models of these methods are not accurate enough.** The performance model behind the threshold based rules and target tracking is actually "QoS violations will not occur when the workload (QPS) is within a certain range". And the parameters of this range are determined based on manual experience. The results in Table 1 prove that it is not effective. In addition, the queuing theory relies too much on theoretical assumptions and is not practical enough, causing its estimated RT to be lower than the true value (hence QoS violations). Note that the workload is known in advance in this experiment, so the QoS violations are only due to an inaccurate performance model.

## 2.3 Challenge

Large-scale web applications auto-scaling faces the following challenges.

**Accurately predict the workload of all web applications.** As discussed in Section 2.1, a significant portion of web applications exhibit a robust period pattern in their QPS (time series). Consequently, the proactive scaling approach based on prediction holds great promise. Nevertheless, the multitude of diverse and complex time-series QPS characteristics generated by real-world applications implies that no single SOTA prediction model will consistently outperform the others, as indicated in Observations 2 and 3 in Section 2.1.

**Construct an effective performance model without adverse impact on online applications.** Due to the large number of applications, it is very expensive to build a complete offline environment for profiling. The performance models of common methods are based on manual experience or statistical mathematics, which cannot effectively guarantee QoS (Section 2.2).

**The QoS requirements of web applications are diverse.** Different applications have different QoS requirements, such as average latency and various percentile tail latencies (TP99, TP999, etc.). Moreover, some applications require a combination of multiple metrics, which increases the difficulty of guaranteeing QoS.

## 3 DESIGN

There are two keys to achieving efficient auto-scaling (guaranteeing QoS and reducing resource cost) in a large-scale enterprise environment. One is robust, reliable and accurate **workload prediction**,

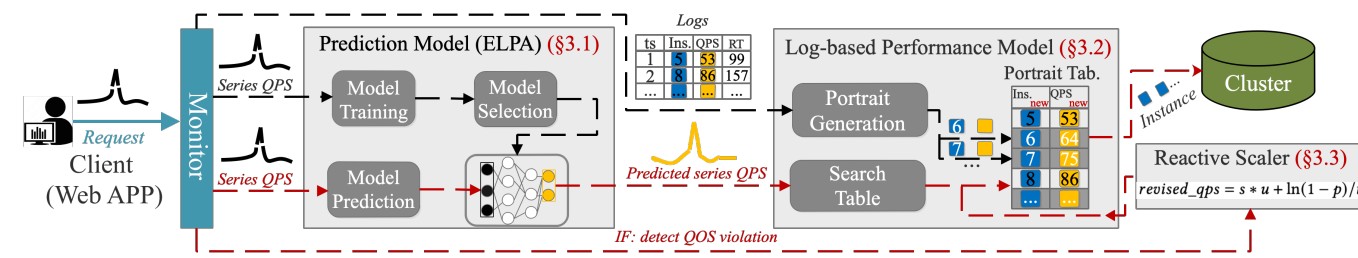

Figure 2: Overview of PASS. The black lines in the upper half figure represent the offline steps. The red lines in the lower half figure represent the online process.

which can adapt to various applications. The second is an effective **performance model** in the online production environment to efficiently ensure QoS with appropriate resources.

The overview of PASS is shown in Figure 2. PASS has an ensemble workload prediction model, named ELPA, for accurately predicting the QPS in real-time (Section 3.1). Then PASS queries the performance model, which is constructed based on historical logs, and predictively scales to the desired number of instances without violating QoS (Section 3.2). In addition, PASS continuously monitors QoS metrics. If a violation is detected resulting from inaccurate workload prediction, PASS quickly revises the prediction result and scales out to the appropriate number of instances to promptly address the violation (Section 3.3).

## 3.1 Workload Prediction

This section introduces an **E**nsemble **L**earning-based **P**rediction **A**lgorithm framework named **ELPA**, which is designed to address the challenges identified with existing prediction algorithms in Section 2.1. The structure of the prediction model is illustrated in Figure 3. For each category of real-time traffic, a corresponding set of online and offline models is employed to provide predictive services. Initially, from a diverse array of online models, we select the one that offers the most accurate prediction for the current time series data distribution. Subsequently, for predicting "mutation features," we opt for the offline prediction model, which excels in learning these features associated with time series traffic. This offline model effectively prediction the "shape" of these "mutation features." Additionally, we have addressed the issue of "traffic amplitude deviation" that can arise when using offline prediction in the context of "mutation features," further enhancing predictive performance. It's worth noting that while "mutation features" constitute a smaller fraction of actual time series data, precise prediction of these features is crucial for maintaining QoS in a production setting. Businesses often grapple with effectively managing these 'mutation features,' which can lead to significant performance degradation or cost implications.

In the following, we will delve into the specifics of online/offline model selection and amplitude calibration within the framework.

*3.1.1 Selection of the Prediction Model in ELPA.* We employ specialized prediction models tailored to each type of time-series traffic data generated by real businesses to ensure optimal predictive performance. The selection process between online and offline models involves evaluating the predictive results of several consecutive cycles to determine which predictive algorithm offers the best performance for the current time-series traffic characteristics. To assess these consecutive cycle outcomes, we employ an exponential weighted average, giving greater weight to predictions closer to

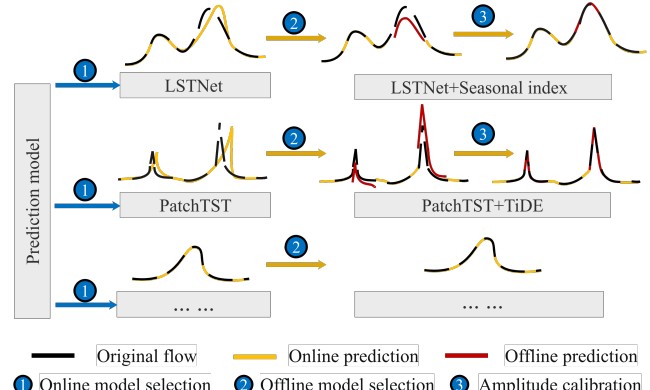

Figure 3: Integration framework of Prediction models (ELPA).

the present moment and gradually decreasing weight for older predictions. We have established an average cumulative index of predictive performance for the evaluation of the current predictive algorithm, as depicted in Equation (1):

$$V_{on} = \frac{\sum_{i=0}^{i=t-1} v_{T-1,i}}{t} \tag{1}$$

$t$ stands for the count of time points within a single period. $v_{T-1,i}$ symbolizes the cumulative metric of single-point predictive performance, used to depict the aggregate value of absolute errors at a specific time point $i$ within each period ranging from $(0, T-1)$ before the present period $T$ (see Equation (2)). As a result, the mean cumulative metric of predictive performance $V$ represents the average of the weighted accumulated absolute errors of all predictive results within the period, serving as an indicator of the predictive efficacy of the algorithm.

$$v_{T-1,i} = \beta * v_{T-2,i} + (1 - \beta) * ABE_{T-1,i} \tag{2}$$

In Equation (2), we have applied the concept of exponential weighted moving average, where the highest weight is assigned to the predictive result closest to the current time point. This approach is founded on our assumption that the traffic features nearest to the current time point most accurately represent the current traffic conditions. $\beta$ denotes the exponential weight. $ABE_{T-1,i}$ signifies the absolute error (absolute error) of the predictive outcome corresponding to a specific predictive algorithm within the current cycle of $T-1$ at time point $i$. Here, $ABE_{T-1,i} = |y_{T-1,i} - \hat{y}_{T-1,i}|$. $y_{T-1,i}$ represents the actual value of traffic at time point $i$ within the cycle of $T-1$. $\hat{y}_{T-1,i}$ denotes the predicted traffic value at time point $i$ within the cycle of $T-1$.

Note that the selection process for both online and offline predictive models relies on the previously mentioned average cumulative

predictive performance metrics ($V_{on}$ and $V_{off}$). The selected online model will handle the bulk of predictive tasks, while the operational details of the offline model will be detailed in Section 3.1.2.

*3.1.2 The operational timing of the offline model in ELPA.* As we discovered in Section 2.1, there are instances when the predictive performance of the online model falls short compared to the offline model, particularly in cases of traffic exhibiting "mutant features". This is primarily due to the offline model's predictive results exhibiting mutation features similar to the actual traffic, which greatly enhances the potential for improving predictive performance. Consequently, it is crucial to devise a mechanism for selecting the appropriate timing for the online/offline model prediction service for the current feature traffic, i.e., determining how to identify these "mutant features".

Specifically, to articulate the "mutant feature", we define the slope of the traffic value between two consecutive points in the time-series traffic data, as depicted in Equation (3). Herein, $y_i$ represents the traffic value at the ith moment, and $x_i$ signifies the timestamp corresponding to the ith moment.

$$K_{(i-1,i)} = |\frac{y_i - y_{i-1}}{x_i - x_{i-1}}| \tag{3}$$

When $K_{(i-1,i)} \geqslant \epsilon$, we characterize the traffic feature at this point as a "mutant feature", and designate moment $i$ as the occurrence of the "mutant feature". On the contrary, we identify it as a more stable traffic feature. In this context, $\epsilon$ is the hyperparameter determining whether it's a "mutant feature". When a specific moment $i$ is classified as a "mutant feature", we employ an offline model to deliver prediction services within the "dirty interval" $[i, i + j]$ (for the definition of "dirty interval", please refer to Section 2.1).

*3.1.3 Amplitude calibration of offline model in ELPA.* The predictive capability of the offline model is totally dependent on the time series data from several previous periods. If the amplitude of the current period undergoes a significant change compared to the amplitudes of past periods, it can result in substantial prediction deviations. Therefore, an amplitude calibration mechanism needs to be designed for the offline model, taking into account the real-time traffic of the day.

Specifically, when the "mutant feature" occurs (at moment $i$), we will determine the calibration multiple $Diff_i$ for the actual traffic value and the offline predicted traffic value at the moment of the "mutant feature" using Equation (4).

$$Diff_i = \frac{\sum_{z=1}^{z=a} TF_{i-z}/PF_{i-z}}{a} \tag{4}$$

In this context, $TF_i$ represents the actual traffic value at the moment $i$, while $PF_i$ signifies the offline predicted traffic value at the same moment. The calculation of the difference multiple should be determined based on the average difference multiple of all points within the time interval $[i-a, i)$. $a$ is a hyperparameter, and its value varies depending on the distinct traffic features. $a$ represents the count of points number within the interval $[i - a, i)$. Therefore, the offline predicted traffic value at the moment $i$ will be recalibrated to $PFC_i = Diff_i * PF_i$.

Note that the amplitude calibration mechanism within the "dirty interval" $[i, i + j]$ aligns with the calibration mechanism at moment

$i$. In other words, if the offline prediction value at the moment $i + 1$ undergoes calibration, the calibration multiple at this point would be $Diff_{i+1}$. Subsequently, the traffic value post-calibration becomes $PFC_{i+1}$.

## 3.2 Log-based Performance Model

---

**Algorithm 1:** Performance Model Construction

**Input:** logs, QoS
**Output:** ins_qps_table

**1** **for** *each (ins_cnt,data) in logs.groupby("ins_cnt")* **do**
**2**    qps_count, violation_count = {}, {}
**3**    **for** *each row in data.itertuples()* **do**
**4**      qps = row["qps"] // cap
**5**      qps_count[qps] += 1
**6**      **if** *data violates QoS* **then**
**7**        violation_count[qps] += 1
**8**      **end**
**9**    **end**
**10**    **Function** QoSGuaranteeRate(*qps*):
**11**      total_count, total_violation = 0, 0
**12**      **for** *each (k,v) in qps_count.items() in ascending order* **do**
**13**        **if** $k > qps$ **then**
**14**          break
**15**        **end**
**16**        total_count += v
**17**        total_violation += violation_count[k]
**18**      **end**
**19**      **return** (total_count-total_violation)/total_count
**20**    lt = sorted(qps_count, key = lambda k:(QoSGuaranteeRate($k$) > $\delta$, k), reverse=True)
**21**    ins_qps_table[ins_cnt] = lt[0] * cap
**22** **end**

---

**Model construction.** Algorithm 1 illustrates how to construct a performance model based on historical logs without offline profiling. The input logs from the monitoring system include information about QPS, the number of instances, and QoS metrics. The input QoS is application-related, and the performance model needs to be re-established when QoS changes. We first aggregate the input logs by the number of instances and traverse all records (line 1-9). Line 4 aggregates QPS by the granularity of "cap" (such as one-thousandth of the maximum QPS) to reduce the number of data (QPS) and algorithm overhead. Lines 5-8 count the occurrence times and the number of QoS violation times. Given some records may be inaccurate resulting from system failures, when we evaluate the guarantee rate of a certain QPS, we not only calculate the current QPS records but also comprehensively take into account all lower QPS records(line 10-19). The sorting rule of line 20 is to prioritize the QoS guarantee rate greater than a given threshold $\delta$, and then sort in descending order by QPS. $\delta$ can be adjusted according to the needs of the application owners (default is 0.99), which measures the tolerance to QoS violations. The closer $\delta$ is to 1, the lower the tolerance. Finally, we assign the first QPS after sorting as the maximum traffic that the current number of instances can handle (line 21).

**Process inaccurate entries.** Performance model tables directly constructed from monitoring logs may be inaccurate. When the

auto-scaling parameters set by the application are unreasonable (such as a large scale-out step or excessive resource redundancy), there may be no monitoring logs for certain instance numbers, or the count may be extremely small. This can lead to missing or inaccurate entries in the initially built table. To solve this, we first ensure that the original table data maintains a non-strict monotonic increase, with empty or lower QPS being replaced by the previous higher QPS. Then, for sections where the number of instances increases but the QPS remains unchanged, we calculate the slope based on the adjacent QPS and update them. For example, if the initialized performance model mapping is $\{5:30, 7:20, 8:60\}$, after using the QPS of instance 5 to replace instance 6 and 7, it becomes $\{5:30, 6:30, 7:30, 8:60\}$. Then, based on the QPS difference between instance 5 and 8, we update the QPS of instance 6 and 7, resulting in $\{5:30, 6:40, 7:50, 8:60\}$. In addition, the performance model is periodically reconstructed using the latest monitoring logs during the low-peak period every night to maintain accuracy.

### 3.3 Hybrid Auto-scaling

---

**Algorithm 2:** Hybrid Auto-scaling

---
**Input:** ins_qps_table, QoS
1 **while** *1* **do**
2    sleep 1 min      // wait new monitor data
3    cur_qps,cur_slo = query_monitor_system()
4    predicted_qps = get_predicted_qps()
5    **if** *cur_slo violates QoS* **then**
      // reactive scaling
6       revised_qps = QT_model(cur_ins_num,cur_slo)
7       qps = max(cur_qps,predicted_qps,revised_qps)
8       scale2ins_num(*max(*search(*qps,ins_qps_table),cur_ins_num+1)*)
9       disable scaling in for a while
10    **end**
11    **else**
      // proactive scaling
12       qps = max(cur_qps,predicted_qps)
13       scale2ins_num(*search(qps,ins_qps_table)*)
14    **end**
15 **end**

---

Not only predictive scaling, PASS also has a reactive strategy grounded in queuing theory to deal with the QoS violations caused by inaccurate workload prediction or sudden load increases in hot events. Our hybrid auto-scaling algorithm is shown in Algorithm 2. PASS monitors the current QPS and Service Level Objective (SLO) of the application and the latest predicted QPS in real time (line 3-4). If a QoS violation is detected, PASS revises the predicted QPS based on the MMs queuing theory model and re-query the performance model to quickly scale out to an appropriate number of instances (lines 6-8). Specifically, the queuing theory model is shown in Equation 5:

$$revised\_qps(M/M/s) = s*u + \ln(1-p)/t \qquad (5)$$

where $s$ represents the current number of instances, $u$ stands for the bottleneck QPS per instance, $p$ denotes the percentile of latency, and $t$ refers to the tail latency at the $p$ percentile (see the

Appendix A.2 for the derivation process, function "search" and "scale2ins_num"). The reason for using queuing theory to estimate QPS instead of latency is that the latency derived from queuing theory tends to be lower than the actual value (as evidenced by the low QoS guarantee rate of queuing theory in Section 2.2), hence the QPS derived from actual latency would be higher than the actual value. We scale out based on the higher QPS to quickly respond to QoS violations and minimize losses. The experiments in Section 4.3 show that this part of resource redundancy does not lead to a lot of waste. line 9 disables scaling in for a period of time in order to prevent newly expanded instances from being scaled in because of system jitter. If no QoS violation is found, PASS performs proactive scaling according to the monitored QPS, predicted QPS, and performance model table (lines 12-13).

## 4 EVALUATION

We implemented PASS with Python (about 3 KLOC) and conducted experiments on our private container platform.

### 4.1 Experiment Setup

**Applications and workloads.** We randomly selected 225 web-related applications from various business lines of our enterprise to verify the accuracy of the prediction model. The historical QPS data of these applications is obtained from the unified online monitoring platform within our enterprise. The data are categorized based on their prediction difficulty into three distinct levels. Among them, 164 are defined as easy, 48 are defined as medium difficulty, and 13 are defined as hard (see the Appendix A.1 for grading standards). We also selected several representative applications for end-to-end evaluation. These applications are back-end services that provide C-side user basic attributes, user behavior query, search and chat functions. We recorded online request traffic and played them back in the offline environment to restore the real online environment. In order to reduce time and resource costs, we sliced the recorded traffic and ignored long-term stable low-peak loads that cannot trigger auto-scaling.

**Comparison baselines.** We compared various types of prediction algorithms and auto-scaling methods. Prediction algorithms include: offline algorithm (Seasonal Index [9], Prophet [41]) and online algorithm (LSTNet [25], PatchTST [32], and TIDE [11]). Note that the online algorithms we have assessed are designed to predict the value at the third time point from the present moment, i.e., horizon equals 3. This is sufficient to meet the demands of predictive scaling. As for offline prediction algorithms, they are capable of predicting time series data for an entire future period in a single operation. Auto-scaling methods include: target tracking [3] and AHPA [49]. Since target tracking can usually scale to the target range faster than the threshold-based rules with the same parameters (Section 2.2 also shows that the performance of target tracking is better), we only compared the target tracking among the commonly used methods. AHPA, which is a SOTA work from Alibaba, has better performance than HPA [31], so we chose it as another comparison object. AHPA is implemented based on the unofficial implementation of RobustPeriod [5] [45] and RobustSTL[6] [44] and

---

[5]https://github.com/ariaghora/robust-period
[6]https://github.com/LeeDoYup/RobustSTL

 

| | Method | Seasonal_index [9] | | Prophet [41] | | LSTNet [25] | | PatchTST [32] | | TIDE [11] | | ELPA | |
|---|---|---|---|---|---|---|---|---|---|---|---|---|---|
| | Metrics | RRSE | CORR | RRSE | CORR | RRSE | CORR | RRSE | CORR | RRSE | CORR | RRSE | CORR |
| Datasets | Metrics_avg | 0.1359 | 0.9934 | 0.3163 | 0.9509 | 0.0801 | 0.9968 | 0.0726 | 0.9973 | 0.1807 | 0.9812 | **0.0651** | **0.9985** |
| (easy) | Winner (rate) | 0% | 0% | 0% | 0% | 3.04% | 1.83% | 3.66% | 2.44% | 0.61% | 0.61% | **92.68%** | **95.12%** |
| Datasets | Metrics_avg | 0.3418 | 0.9588 | 0.4418 | 0.8936 | 0.1853 | 0.9704 | 0.1576 | 0.9826 | 0.3089 | 0.9489 | **0.1311** | **0.9873** |
| (medium) | Winner (rate) | 0% | 0% | 0% | 0% | 2.08% | 2.08% | 6.25% | 4.17 | 0% | 0% | **91.67%** | **93.75%** |
| Datasets | Metrics_avg | 0.6009 | 0.8514 | 0.6063 | 0.7996 | 0.2848 | 0.9147 | 0.2452 | 0.9495 | 0.4527 | 0.8788 | **0.1903** | **0.9571** |
| (hard) | Winner (rate) | 0% | 0% | 0% | 0% | 7.69% | 0% | 0% | 7.69% | 0% | 0% | **92.3%** | **92.3%** |

**Table 2: This is an accuracy summary of SOTA methods on various datasets, divided into three prediction difficulty levels. Each row and column compare the results of all methods at a specific level's datasets and for a specific metric, respectively. Boldface highlights the best result per metric.** *Metrics_avg* **represents the average prediction outcomes at current difficulty levels' datasets, while** *Winner* **shows the proportion where a method outperforms others in accuracy, evaluated per dataset type.**

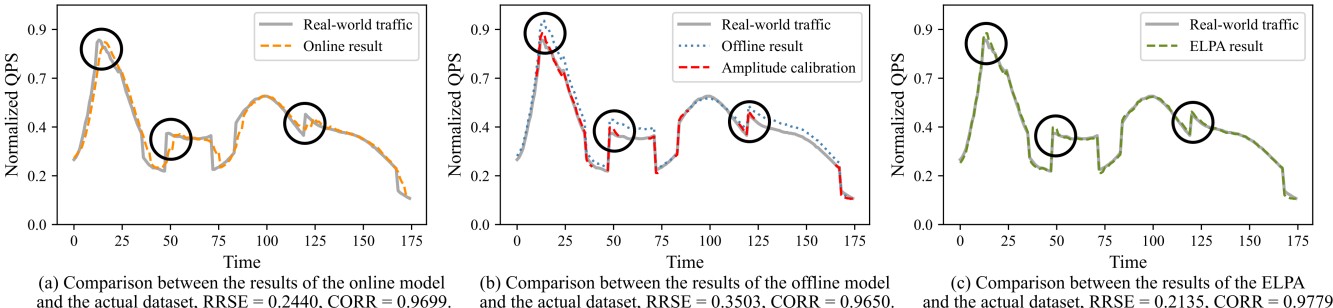

(a) Comparison between the results of the online model and the actual dataset, RRSE = 0.2440, CORR = 0.9699.

(b) Comparison between the results of the offline model and the actual dataset, RRSE = 0.3503, CORR = 0.9650.

(c) Comparison between the results of the ELPA and the actual dataset, RRSE = 0.2135, CORR = 0.9779.

**Figure 4: Exhibit of prediction instantiation for the online model, offline model (inclusive of amplitude calibration), and ELPA.**

the description of the paper. The other algorithms are the official open-source implementation.

## 4.2 Evaluation of the Ensemble Learning-based Prediction Algorithm (ELPA)

In this section, we conduct an evaluation of ELPA, alongside five other widely employed prediction algorithms, as detailed in Section 4.1. These methodologies find practical application in time series data forecasting within prominent cloud service providers. For instance, Prophet has been adopted within Facebook's forecasting contexts [41], underscoring their relevance in enterprise-level scenarios. Moreover, the ELPA framework exhibits flexibility in integrating SOTA prediction algorithms, thereby enhancing their predictive capabilities without being limited to the algorithms mentioned earlier[7]. Subsequently, our assessment will center on the accuracy of these prediction algorithms (see Section 4.2.1) and their robustness (see Section 4.2.2).

*4.2.1 Overall evaluation of ELPA's accuracy.* Table 2 presents the comparative prediction accuracy[8] performance of various prediction algorithms. It is evident that the ELPA framework outperforms individual prediction algorithms across most metrics. This superior performance can be attributed to ELPA's implementation of a set of optimized rules for selecting the most suitable online/offline combination and employing amplitude calibration to predict specific time series data.

However, it is worth noting that in specific datasets, certain online models exhibit superior performance. For instance, in the 'easy' datasets, LSTNet and PatchTST respectively win in 3.04% and 3.66% of the dataset types (RRSE). This variation can be attributed

---

[7]The ELPA framework is adaptable to integrate any SOTA prediction algorithm.
[8]Please refer to the Appendix A.5 for the RRSE and CORR evaluation metrics.

to minor shifts in data distribution, causing the best online model to vary between training and testing datasets. For example, while PatchTST performs as the top prediction algorithm in the testing data (in real-world scenarios), ELPA, informed by the training data, selects LSTNet, the highest-performing prediction algorithm, for predicting the testing data. Nonetheless, in these dataset categories, the prediction results of ELPA deviate by a maximum of only 0.5% when compared to the top-performing models, indicating nearly identical prediction performance. Thus, we have confidence in ELPA's capability to consistently deliver exceptional prediction performance across various scenarios.

**Insight 1: The ELPA prediction framework leverages the advantages of both online and offline models, consistently achieving superior prediction accuracy in most scenarios. Even in infrequent scenarios where a single prediction model outperforms, ELPA consistently maintains a prediction accuracy remarkably close to it.**

*4.2.2 The robustness evaluation of ELPA.* In Section 2.1, we have come to recognize the significance of accurately forecasting "mutation features" to ensure QoS. Figure 4 provides an illustrative example of ELPA's approach, where it combines online and offline models with an amplitude adjustment for the offline model. It's important to note that "mutation features" can manifest in various types of real-time sequential data. Due to space constraints, we've chosen one representative time series dataset to highlight ELPA's robust predictive capabilities.

In Figure 4(a), a comparison between the predictions made by the online model (PatchTST) and the actual data is presented. While the online model exhibits high prediction accuracy in most cases, a significant lag issue is observed in predicting "mutation features" (highlighted in a black circle in Figure 4). This lag could potentially hinder the provision of predictive performance. In Figure 4(b), the

| | Scenario 1 TP99 ≤ 40, 3h | | Scenario 2 TP999 ≤ 80, 4h | | Scenario 3 TP999 ≤ 40, 4h | | Scenario 4 TP999 ≤ 30, 4h | | Scenario 5 TP999 ≤ 95, 6h | | Scenario 6 TP99 ≤ 45 & TP999 ≤ 100, 4h | | Average | |
|---|---|---|---|---|---|---|---|---|---|---|---|---|---|---|
| Target Tracking | 94.44% | 27.35 | 80.83% | 32.18 | 88.33% | 24.68 | 99.17% | 25.40 | 93.89% | 36.03 | 69.44% | 29.28 | 87.68% | 29.15 |
| AHPA | 87.78% | **23.42** | 80.83% | 30.68 | 86.67% | **19.82** | **100.00%** | 20.00 | 92.22% | 38.75 | 67.50% | 26.67 | 85.83% | 26.56 |
| PASS | **97.78%** | 24.53 | **88.75%** | **29.60** | **91.25%** | 19.83 | **100.00%** | **12.00** | **94.17%** | **33.63** | **82.78%** | **25.55** | **92.46%** | **24.19** |

**Table 3: The end-to-end performance of three auto-scaling methods under different QoS and test duration (hour) scenarios. In each scenario, the first column is the QoS guarantee rate, and the second column is the resource cost(instance num * hours). Boldface highlights the best result per metric.**

predictive performance of the offline model (Seasonal Index model) is depicted by the blue dashed line. Despite a noticeable prediction discrepancy with the actual data in most cases, it effectively anticipates "mutation features" in a timely manner, mitigating the problem of prediction lag. The red dashed line in Figure 4(b) shows the result of ELPA's amplitude calibration at the "mutation features" of the offline model, successfully predicting these features. Finally, in Figure 4(c), ELPA's outcomes are presented, which integrates the online model and the calibrated offline model. It's evident that ELPA's prediction accuracy and its ability to predict "mutation features" both demonstrate exceptional predictive performance.

**Insight 2: The offline model, while proficient at capturing the data's "mutation features," shows a notable discrepancy in its predictive results. Conversely, the online model encounters challenges in effectively forecasting these "mutation features." However, the ELPA framework, through the amalgamation of both models and the application of amplitude calibration, showcases remarkable robustness.**

### 4.3 End-to-end Performance

**Metrics.** We evaluate the effect of the auto-scaling method from two aspects. The *QoS guarantee rate* measures the length of time an application violates QoS, which is computed by taking the ratio of the duration for which QoS is guaranteed to the total time length. The *Resource cost* is determined by integrating the quantity of instances over a given time period, measured in hours.

The results of the end-to-end experiment are shown in Table 3 (see Appendix A.4 for detailed monitoring data during the test process, including QPS series, instance count, and QoS indicators such as TP99, TP999 tail latency). Each test scenario provides QoS indicators, test duration, and the QoS guarantee rate and resource cost of the three methods.

**Insight 3: The performance model of PASS is accurate and effective.** PASS achieves the highest QoS guarantee rate in all application test scenarios. Compared with target tracking and AHPA, the average QoS guarantee rate has increased by 5.54% and 7.71%. It is even more obvious in scenario 6 of multiple QoS indicators, where our QoS guarantee rate has increased by 19.21% and 22.64%, respectively. PASS also has the lowest average resource cost in all scenarios. The average resource cost is reduced by 8.91% compared to AHPA and 17.02% compared to target tracking. Our resource cost is reduced by up to 40% and 52.76% in scenario 4. In only two scenarios, the resource cost of PASS is slightly higher than AHPA, and in all other tests, our resource cost is the lowest.

Note that the application instance is started without pre-warming (For example, the database service establishes a connection in advance, loads the metadata required at runtime into cache, etc.), so even if the instance is scaled out in advance, a large number of cold queries at the beginning will still cause a sudden increase in tail latency. If pre-warming is added to the application instance startup logic, our QoS guarantee rate will be further improved.

## 5 RELATED WORK

In Section 2.2 we introduced some auto-scaling methods, and we provide several other related work in this section.

**AI for time series.** Multivariate time series data pervade our daily lives, encompassing aspects such as cloud service time series load, the energy output of solar power plants, electricity consumption, and traffic congestion [27, 46]. Users frequently express interest in analyzing or forecasting new trends or potential risks, leveraging historical observations from time series signals. This paper primarily focuses on the load prediction task, which provides crucial guidance for auto-scaling [10, 23, 32]. Moreover, prevailing research typically revolves around Time Series Anomaly Detection [6, 26, 35], Time Series Classification [16, 29, 40], and other facets of Time Series Analysis [30, 47, 48].

**Hybrid auto-scaling approaches.** Due to the pros and cons of individual methods or models, some works have proposed hybrid scaling approaches. Pereira [34] proposed a horizontal scaling method that combines threshold-based and time-series prediction. It uses monitoring data for reactive scaling and proactively scales based on time-series prediction results. Jamshidi [22] proposed a method that combines machine learning and rule-based scaling, which can automatically design and adjust rules dynamically without human involvement. Gambi [15] proposed a hybrid approach that combines machine learning and queuing theory. During the process of training the machine learning model, queuing theory is used as a substitute to address the problem of QoS violations caused by slow convergence.

## 6 CONCLUSION

We have presented PASS, a predictive auto-scaling system for large-scale online web applications in enterprises. We designed an integrated prediction framework namely ELPA to cope with complex and diverse application workloads, and constructed a performance model through logs for predictive scaling. In addition, we also implemented a reactive scaling strategy based on queuing theory to quickly respond to QoS violations. Under a diverse range of enterprise applications and under real workloads, ELPA outperforms existing single prediction models in 90%+ of web service scenarios. Notably, in scenarios featuring "mutant features," ELPA demonstrates a remarkable enhancement in predictive accuracy, reaching an increase of up to 36.1% when compared to the second-best prediction model. Employing ELPA, PASS effectively lowers average resource consumption by 8.91% compared to AHPA while achieving a notable improvement in QoS assurance, with a rate increase of up to 22.64%.

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

# A APPENDIX

## A.1 Identifying Periodic Patterns

We regularly analyze time-series data representing the traffic load of web applications running on a genuine platform. This data is collected at weekly intervals, representing the window size. By aggregating traffic data over multiple consecutive windows, we ascertain whether these applications exhibit any recurring patterns or periodicity. The specific method of determination involves calculating the Pearson correlation coefficient and the mean of any two windows. Please refer to Equation 6.

$$\rho = \frac{\sum (X_i - \bar{X})(Y_i - \bar{Y})}{\sqrt{\sum (X_i - \bar{X})^2 \sum (Y_i - \bar{Y})^2}} \tag{6}$$

$X$ and $Y$ represent time series data for two consecutive weeks. $X_i$ represents the QPS value at time $i$ during the current week, while $\bar{X}$ and $\bar{Y}$ denote the means of $X$ and $Y$, respectively.

The Pearson correlation coefficient is a statistical measure used to assess the strength and direction of the linear relationship between two continuous variables. When analyzing time series data, if the correlation coefficient exceeds 0.8 and the difference in the means is within 5%, the application is considered to exhibit strong periodicity. If the correlation coefficient falls between 0.5 and 0.8, and the difference in the means is within 10%, the application is regarded as having weak periodicity [17]. Otherwise, the application is considered to lack periodicity.

In Section 4.1, we classified the prediction dataset into three distinct levels of complexity: easy, medium, and hard. This classification process relied on the computation of correlation, using the Pearson Correlation Coefficient (defined in Equation 6), on the QPS time series data from the preceding two weeks. When the results demonstrated a robust correlation, the first half portion was designated as "easy," and the latter half portion as "medium." In cases where the results exhibited a weak correlation, they were categorized as "hard." If the outcomes indicated a complete absence of correlation, it signaled that forecasting was unnecessary due to the lack of underlying periodic patterns.

## A.2 The Derivation Process of Equation 5

Assume that the time interval between request arrivals follows an exponential distribution with parameter $\lambda$, the service times have an exponential distribution with parameter $u$, and the number of servers is s. Then in M/M/s model, the request's sojourn time T in the system follows an exponential distribution with parameters $s \cdot u - \lambda$, and its probability density function is:

$$f(t) = \begin{cases} (s \cdot u - \lambda)e^{-(s \cdot u - \lambda)t} & \text{for } t > 0, \\ 0 & \text{otherwise.} \end{cases}$$

Therefore, the cumulative distribution function is:

$$P\{T \le t\} = 1 - e^{-(s \cdot u - \lambda)t}, \quad t \ge 0$$

Let $P\{T \le t\} = p$ ,which is the percentile of tail latency, we get:

$$1 - e^{-(s \cdot u - \lambda)t} = p$$
$$\ln(1 - p) = -(s \cdot u - \lambda)t$$
$$\ln(1 - p)/t = -s \cdot u + \lambda$$
$$\lambda = s \cdot u + \ln(1 - p)/t$$

Hence, the QPS estimated by M/M/s queuing model based on the number of servers and $p$ percentile tail latency $t$ is:

$$qps = s * u + \ln(1 - p)/t$$

## A.3 Functions used in Hybrid Auto-scaling

Algorithm 3 provides the function "search" and "scale2ins_num" used in Algorithm 2. The function "search" (line 1-7) queries the performance model to determine how many instances are needed to handle the incoming QPS traffic. The function scale2ins_num (lines 8-15) is used to scale to the input number of instances. Note that newly started instances require a certain amount of warm-up time, scaling out will sleep for a while (line 11). There is no need for scaling in to sleep as it takes effect almost in real time.

---

**Algorithm 3:** Functions used in Hybrid Auto-scaling

1   **Function** search(*QPS,ins_qps_table*):
2     **for** *ins_num,qps in ins_qps_table in ascending order* **do**
3       **if** *qps ≥ QPS* **then**
4         **return** ins_num
5       **end**
6     **end**
7     **return** MAX_INS_NUM
8   **Function** scale2ins_num(*num*):
9     **if** *num > cur_ins_num* **then**
10       scale_out(num - cur_ins_num)
11       sleep(warmup_time)
12     **end**
13     **else if** *num < cur_ins_num* **then**
14       scale_in(cur_ins_num - num)
15     **end**

---

## A.4 The Monitoring Data during End-to-end Performance Test

Figure 5 and Figure 6 provide detailed monitoring data of several scenarios in 4.3, including QPS series, instance count and QoS metrics. The QPS curve graph includes the actual request traffic curve (with data normalized to remove sensitive information) and the predicted curve (AHPA and PASS only because target tracking is a reactive method that does not have prediction models). In the tail latency curve graph, the blue line represents the tail latency (logarithmic transformation has been applied for better observation). Points above the pink line indicate QoS violations. Note that the relatively high spurt is because the application instance is started without pre-warming.

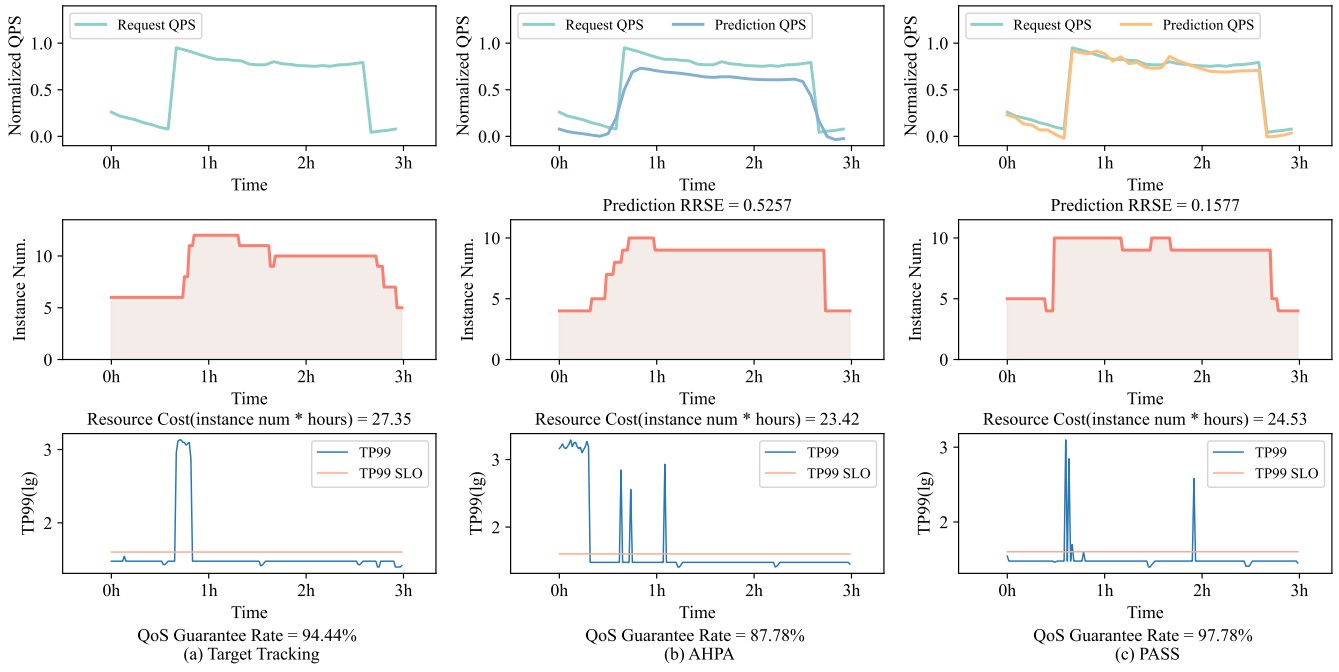

Figure 5: The comprehensive monitoring data for Scenario 1 encompasses QPS series, instance count, and QoS metrics.

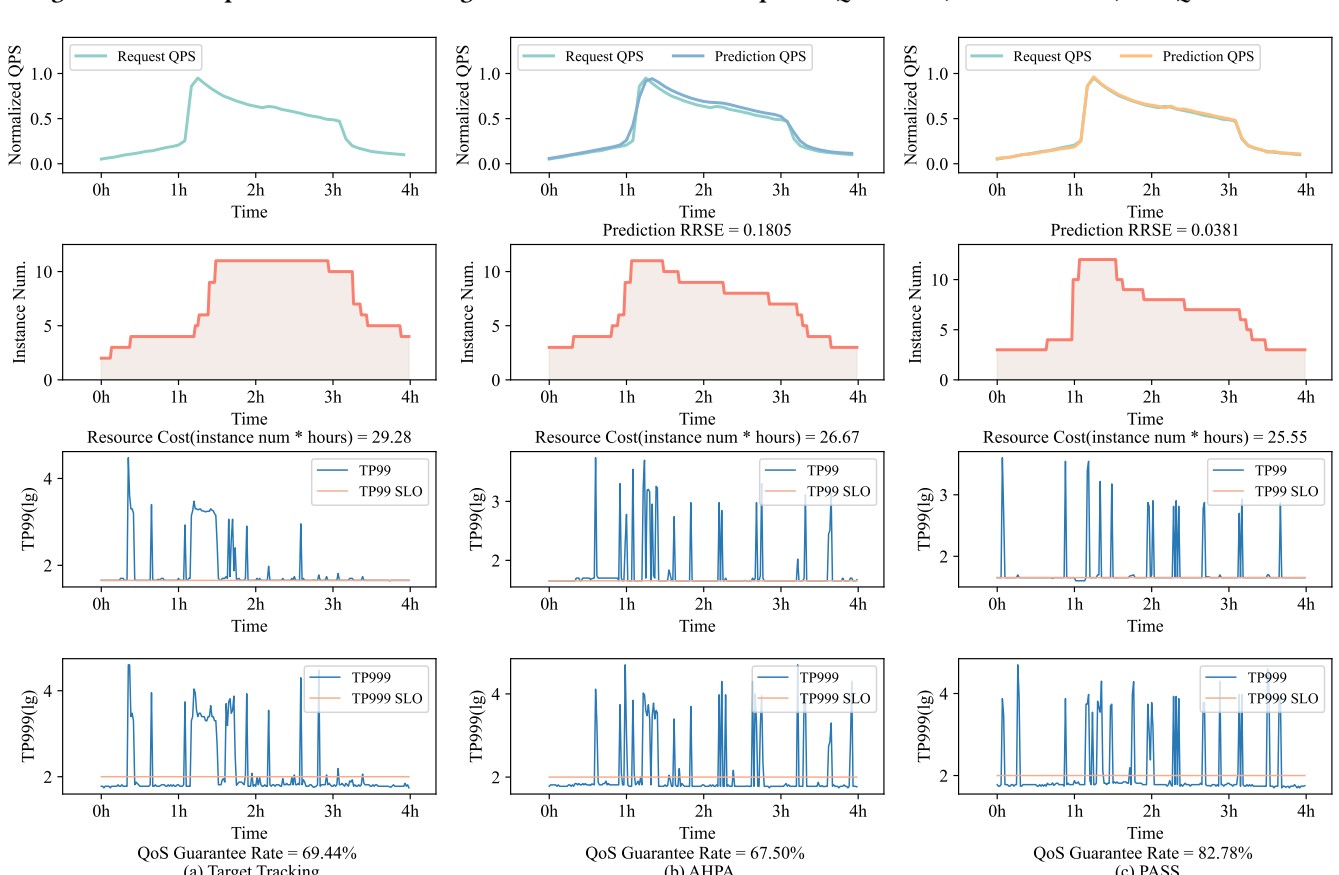

Figure 6: The comprehensive monitoring data for Scenario 6 encompasses QPS series, instance count, and QoS metrics.

## A.5 The evaluation metrics for prediction algorithms, RRSE and CORR.

*A.5.1 RRSE (Relative Root Mean Square Error):* RRSE is a metric used to assess the accuracy of a time series forecasting model. It measures the relative error between the forecasted values and the actual values in the time series data. A lower RRSE indicates a more accurate forecast, as it suggests that the forecasted values are closer to the actual values. The formula for RRSE is typically as follows (Equation 7):

$$RRSE = \frac{\sqrt{\frac{\sum_{t=1}^{n}(Y_t - \hat{Y}_t)^2}{n}}}{\sqrt{\frac{\sum_{t=1}^{n}(Y_t - \bar{Y})^2}{n}}} \tag{7}$$

where $Y_t$ is the actual value at time $t$, $\hat{Y}_t$ is the forecasted (predicted) value at time $t$, $\bar{Y}$ is the mean of the actual values, $n$ is the number of data points in the time series.

*A.5.2 CORR (Correlation):* Correlation is a statistical measure that quantifies the strength and direction of a linear relationship between two variables. In the context of time series forecasting, it is used to assess how well the forecasted values correlate with the actual values. When the CORR value approaches 1, it signifies a robust positive linear correlation between the forecasted values and the observed actual values. This implies that as the actual values experience an increase, the forecasted values exhibit a concomitant increase in a coherent fashion. Within the domain of time series forecasting, this phenomenon is typically construed as a favorable indicator, indicative of a model's robust predictive prowess. The formula for RRSE is typically as follows (Equation 8):

$$CORR = \frac{\sum_{t=1}^{n}(Y_t - \bar{Y})(\hat{Y}_t - \bar{\hat{Y}})}{\sqrt{\sum_{t=1}^{n}(Y_t - \bar{Y})^2 \sum_{t=1}^{n}(\hat{Y}_t - \bar{\hat{Y}})^2}} \tag{8}$$

Where $Y_t$ is the actual value at time t, $\hat{Y}_t$ is the forecasted (predicted) value at time $t$, $\bar{Y}$ is the mean of the actual values, $\bar{\hat{Y}}$ is the mean of the forecasted (predicted) values, $n$ is the number of data points in the time series.

