# OpenReview forum: "PASS: Predictive Auto-Scaling System for Large-scale Enterprise Web Applications"
_ACM.org/TheWebConf/2024/Conference — TheWebConf24 Oral_

### Official Review · Reviewer_5vTo · 2023-11-05

**Novelty:** 5
**Technical Quality:** 5

**Review:**

Quality
Overall, this paper is well-written with a clear presentation.
The step-by-step exposition excels in clarity, and the method is well-motivated.
A concern lies in the evaluation. You have combined two models, and your baselines for end-to-end auto-scaling are based on a single model. The evaluation is slightly unfair as they have different levels of complexity.
Moreover, I'm afraid of the overhead incurred by PASS.

Originality:
Combining prediction models with performance models for enterprise-level web application management seems novel.
Though the ensemble learning-based algorithm is not a big innovation and log-based performance models are well-known, the whole idea still adds to the robustness and specificity of the solution from a global view.

Significance:
The topic is very close to this conference's scope. The authors make a compelling case for the economic and performance benefits of their system. The reported improvements in QoS assurance and resource cost reduction have far-reaching implications for the efficiency and profitability of enterprises.

Cons:

1. While the paper is well-written, it might have benefitted from a discussion section where the broader implications of the research could be explored.
2. The paper could provide more details on the scalability and limitations of PASS to give a better understanding of its applicability in varying contexts.
3. The evaluation is not sound enough. I would like to see an overhead comparison and the combination of your prediction algorithm with other SOTA auto-scaling methods. I assume you are the first to combine the two parts, prediction and auto-scaling. If not, please highlight it; otherwise, I'd like to see an evaluation of auto-scaling methods empowered by prediction models.

**Questions:**

1. Can you provide the efficiency evaluation between  PASS and other auto-scaling methods? Will PASS burden the whole process?
2. Will you make the data and code public upon acceptance? If not, why?
3. Have you already implemented PASS in real-world companies? If not, what's the bottleneck to put it into practice?

**Reviewer Confidence:**

3: The reviewer is confident but not certain that the evaluation is correct

**Scope:**

4: The work is relevant to the Web and to the track, and is of broad interest to the community

---

### Official Review · Reviewer_cMey · 2023-11-13

**Novelty:** 4
**Technical Quality:** 5

**Review:**

\textbf{Summary}
- The paper explored the well-known problem of predictive auto-scaling. The novelty resides mostly in the thorough evaluation of PASS in more than 200 enterprise web services.

\textbf{Strength}
- The paper is clear and well-written
- The paper motivates quite well how certain types of prediction algorithms perform better under different scenarios (namely the advantages/disadvantages of time series and offline prediction algorithms)
- Evaluation is clear

\textbf{Weakness}
- The paper focuses on the limitation of offline profiling, although true, there has been a significant amount of work that does online profiles, or like in this work, leverage historical logs. E.g. Delimitrou et al., Quasar: Resource-Efficient and QoS-Aware Cluster Management
- In Table I'm surprised by the poor results from Prophet which in other works has good performance. Moreover, by looking at the RRSE and CORR I can see that ELPA, PathTST, and LSNET have very similar values, however, the winning rate of ELPA is always above 90%, this means that although winning, in a practical sense there should be none (or minor) QoS effect of using any of the three approaches. It'd have been interesting to have used this prediction to make autoscaling decisions.

Small note: I think it is time to stop saying the 'rapid growth of cloud computing'. The technology has more than 20 years, it is quite mature and well established.

**Questions:**

Please refer to the review. Mostly about my comment on table 2.

**Reviewer Confidence:**

3: The reviewer is confident but not certain that the evaluation is correct

**Scope:**

3: The work is somewhat relevant to the Web and to the track, and is of narrow interest to a sub-community

---

### Official Review · Reviewer_aRRV · 2023-11-23

**Novelty:** 5
**Technical Quality:** 5

**Review:**

Thank you for your submission!

This work presents PASS, an approach for predicting application QPS needs, then incorporating those needs into an autoscaling system that attempts to satisfy the system QoS needs and minimize the duration of violations. The authors outline the limitations of approaches commonly used in enterprise settings today, motivate their design with these limitations, and provide an evaluation which compares the accuracy of the predictions to other approaches, and finally show that the system is able to achieve improved QoS results at modest (or no) cost.

The direction of the overall argument was clear overall, but many of the details were difficult to follow, either due to shifting terminology (mutation features became mutant features), repeated statement of the same arguments with additional details each time (especially across sections 2 and 3, where many of the definitions provided in 3 would have clarified 2 significantly).

While each component of the predictive components of the system (section 3) is well motivated by examples from the data, the components themselves appear like they would have high risk of overfitting — especially the amplitude calibration — since the mutant features may vary significantly across time periods. The importance of each competently would be greatly strengthened by a component analysis which shows the value of each step, e.g. how accurate would the system be if it only did the only training? How important is the amplitude matching? What are the impacts on both the intermediate results (the prediction accuracy) and the resulting QoS behavior.

Exactly what is being evaluated is not clear. Are these showing the RRSE/CORR values across all iterations of the system? Since the selected approach might vary from time window to time window, is there a warm up period where the system is less accurate? How long are the time windows used? How many of them are present here? Is the system looking at hours? Days? Weeks? Not knowing such features of the evaluation makes it difficult to understand the context of the evaluation.

Pros:
- Does a nice job of outlining challenges with diverse autoscaling systems
- Addresses the short falls in the design of the system
- Demonstrates an improvement to QoS compliance time, with modest impact to cost

Cons:
- Many of the scaling seems at high risk of overfitting
- Not clear that the evaluation (esp 4.3) is communicating the predictive accuracy and in what context.
- No component analysis showing the value of each process (e.g. maybe the online prediction is doing all the work, etc.)

Detailed Comments:

- Early on here, I do not have a clear understanding of what exactly is meant by "mutation features" — and the definition comes much later in section 3. A forward pointer could help with clarity.

- Table 1 what's the actual workload we are looking at here? It's difficult to understand what the meaning of these resource costs is without more information. Be sure to clearly define abbreviations from the tables (some of these, eg. TP999 appear later in the text).

- 2.2 - I don't follow what the QoS guarantee actually means. Is there a formal definition? Is this like a duty cycle, describing the total % of the time that the QoS thresholds are satisfied? In all cases, it seems like the take away is that the QoS requirements are not frequently met.

- Some of the explanations of the inadequacies of existing methods lacks some rigor: "In addition, the queuing theory relies too much on theoretical assumptions and is not practical enough, causing its estimated RT to be lower than the true value (hence QoS violations)." -> Should tell us what assumptions, how they impact the system performance.

- 3.1.1 - Definition of V, V_on, V_off is confusing — explaining the on/off portion earlier in the section would help to avoid confusion.

- Im a little confused about how we look back at the previous windows. For example y_{t-1,i} makese sense, but what is y_{t-2, i} — is that the predicted value with last iteration for time i in the t'th window? Does that mean we use last iterations predicted value for the current time, or a "matching" time i in the previous window?

- 3.1.2 -  Mutant feature definition seems like it may introduce significant noise — since steep slopes could occur in  short time periods - e.g from minute to minute, modest increases could exceed this threshold.

- Both components feel like we fit a model, then we scale it so it fits exactly — isn't that likely to introduce severe overfitting? e.g. why do we expect the amplitude scaling to work in general and apply to future time windows?

- 4.2 - Im unclear what this is showing: are we watching the offline model make adjustments to fit the data it has already seen — but that does not tell us that it will do well when seeing new such "mutant points" in the future — e.g is this any good at prediction of future values. It's possible this is showing the performance on new data, but it is not clear from the text.

- 4.3 -  the Qos matching did improve, which suggests that overall the system performs well, which is ultimately convincing.

**Questions:**

How do we know the component contributions of the system? E.g. How necessary is the offline model, the adaptive scaling? While they are well motivated, they lack individual analysis on the data.

Can additional data on the scope of the real world data used to evaluate the system be shared? E.g duration of the data used evaluated, the number of iterations and time windows?

**Reviewer Confidence:**

2: The reviewer is willing to defend the evaluation, but it is likely that the reviewer did not understand parts of the paper

**Scope:**

4: The work is relevant to the Web and to the track, and is of broad interest to the community

---

### Official Review · Reviewer_6Wec · 2023-11-23

**Novelty:** 4
**Technical Quality:** 5

**Review:**

Significance/Pros:

Autoscaling and upholding Quality of Service (QoS) are paramount for large-scale providers, significantly influencing application performance. This concern not only leads to financial losses but also raises performance-related apprehensions. Hence, the issue tackled in this paper holds immense importance.

The solution appears highly promising, and the model results and outcomes are very convincing.

Adequate technical details are presented to substantiate the design, methodology, and formulation of the prediction model.

Cons:
It seems the manuscript lacks a comprehensive understanding of the literature and references to support various arguments.
A substantial amount of research has been conducted regarding prediction-based autoscaling in large-scale applications and systems. Several established cloud providers have adopted mature solutions in this area. Expanding on this literature upfront would be advantageous to strengthen any arguments.

Their claims lack support from existing literature. For example “ AI-driven auto-scaling methods like reinforcement learning are not universally applicable due to their potential adverse impact on the online applications” Evidence is missing.

The authors have outlined two challenges, yet they lack sufficient recent evidence/publications to support their summary. Additionally, they haven't identified any open gaps in the existing literature or current work.

**Questions:**

It seems the manuscript lacks a comprehensive understanding of the literature and references to support various arguments.
A substantial amount of research has been conducted regarding prediction-based autoscaling in large-scale applications and systems. Several established cloud providers have adopted mature solutions in this area. Expanding on this literature upfront would be advantageous to strengthen any arguments.

A few claims lack support from existing literature. For example “ AI-driven auto-scaling methods like reinforcement learning are not universally applicable due to their potential adverse impact on the online applications” Evidence is missing.

The authors have outlined two challenges, yet they lack sufficient recent evidence/publications to support their summary. Additionally, they haven't identified any open gaps in the existing literature or current work.

**Reviewer Confidence:**

3: The reviewer is confident but not certain that the evaluation is correct

**Scope:**

4: The work is relevant to the Web and to the track, and is of broad interest to the community

---

### Official Review · Reviewer_1yav · 2023-11-25

**Novelty:** 7
**Technical Quality:** 7

**Review:**

Autoscaling is a widely-studied problem as it is directly related to the user experiences and operation cost of the services. Although various studies have been done, it is still an important problem as small improvements can reduce operating costs by scale and single failure points can still harm the user experience severely. PASS successfully address such challenges of autoscaling in large-scale enterprise web applications by proposing a practical system design that improves autoscaling performance by utilizing future workload predictions while providing various fallback designs.

The design is pretty convincing. And I expect it to work well in practice.


Pros

-Mixture of online and offline models for accurate time-series prediction of real time workload QPS. Offline models complement the online models in the presence of “mutant features” where unusual workload spikes are expected. Also, the amplitude calibration to complement the limitations of offline models that cannot take account of near past workload data.

-Series of components that ensure better prediction and reasonable fallback scaling policies in case of malfunction from the other components. The end-to-end design components of PASS seems to be capable of handling practical large-scale enterprise web applications.

Cons

-The detection of mutant features are highly dependent toward heuristic threshold.

-The availability of the actual design codes or the collected data used for training of models and validation through evaluations.


Comments:

You should cite recent work on microservice scaling, such as FIRM [OSDI20] and GRAF [CoNEXT21].

**Questions:**

-What causes the tail latency spikes that violates SLO during the operation of PASS? Figure 6 provided in Appendix shows the normalized request QPS of scenario 6. It is surprising how well PASS handles the “mutant features”, even beyond the level I expected. However, it is also noticeable that majority of SLO violations happen where there are no “mutant features” and in rather smooth periods that look easy. How does such a phenomenon take place while operating PASS?

**Ethics Review Description:**

No issues

**Reviewer Confidence:**

4: The reviewer is certain that the evaluation is correct and very familiar with the relevant literature

**Scope:**

4: The work is relevant to the Web and to the track, and is of broad interest to the community

---

### Decision · Program_Chairs · 2024-01-22

**Decision:**

Accept (Oral)

**Comment:**

This paper focuses on predictive autoscaling the infrastructure of a web application. The topic is of high relevance to this community. Reviewers have praised the following aspects: motivation, design, mixture of online+offline models, originality. However, reviewers have raised the following as areas for improvement: mitigating overfitting with a 92-8 data split, citing RW (academic and industry), discussion of implications, and some minor issues (e.g. inconsistent terminology). The authors' rebuttal is reasonable and reassuring (e.g. results from ablation experiments). Overall, I recommend this paper to be accepted due to its high relevance and well executed study.